# Preliminary Analysis of the Salt-Tolerance Mechanisms of Different Varieties of Dandelion (*Taraxacum mongolicum* Hand.-Mazz.) Under Salt Stress

**DOI:** 10.3390/cimb47060449

**Published:** 2025-06-11

**Authors:** Wei Feng, Ran Meng, Yue Chen, Zhaojia Li, Xuelin Lu, Xiuping Wang, Zhe Wu

**Affiliations:** Institute of Coastal Agriculture, Hebei Academy of Agriculture and Forestry Sciences, Tangshan 063299, China; fengwei522106@163.com (W.F.); yoki_meng@163.com (R.M.); xvclaire@163.com (Y.C.); tofriendzhaojia@163.com (Z.L.); nkslxl@163.com (X.L.); bhswxp@163.com (X.W.)

**Keywords:** salt-response, genes, salt-tolerant plant, glycoside hydrolase, adenylate esterases, P450 cytochrome

## Abstract

Soil salinization hinders plant growth and agricultural production, so breeding salt-tolerant crops is an economical way to exploit saline–alkali soils. However, the specific metabolites and associated pathways involved in salt tolerance of the dandelion have not been clearly elucidated so far. Here, we compared the transcriptome and metabolome responses of 0.7% NaCl-stressed dandelion ‘BINPU2’ (variety A) and ‘TANGHAI’ (variety B). Our results showed that 222 significantly altered metabolites mainly enriched in arginine biosynthesis and pyruvate metabolism according to a KEGG database analysis in variety A, while 147 differential metabolites were predominantly enriched in galactose metabolism and the pentose phosphate pathway in variety B. The transcriptome data indicated that the differentially expressed genes (DEGs) in variety A were linked to secondary metabolite biosynthesis, phenylpropanoid biosynthesis, and photosynthesis–antenna proteins. Additionally, KEGG annotations revealed the DEGs had functions assigned to general function prediction only, post-translation modification, protein turnover, chaperones, and signal transduction mechanisms in variety A. By contrast, the DEGs had functions assigned to variety B as plant–pathogen interactions, phenylpropanoid biosynthesis, and photosynthesis–antenna proteins, including general function prediction, signal transduction mechanisms, and secondary metabolite biosynthesis from the KOG database functional annotation. Furthermore, 181 and 162 transcription factors (TFs) expressed under saline stress conditions specifically were detected between varieties A and B, respectively, representing 36 and 37 TF families. Metabolomics combined with transcriptomics revealed that salt stress induced substantial changes in terpenoid metabolites, ubiquinone biosynthesis metabolites, and pyruvate metabolites, mediated by key enzymes from the glycoside hydrolase family, adenylate esterases family, and P450 cytochrome family at the mRNA and/or metabolite levels. These results may uncover the potential salt-response mechanisms in different dandelion varieties, providing insights for breeding salt-tolerant crop plants suitable for saline–alkali land cultivation.

## 1. Introduction

There are about 100 million hectares of saline soils in China, which seriously limits agriculture development [1]. Nevertheless, mild saline stress could enhance osmoprotectants and secondary metabolite contents improving product quality, especially in medicinal species like *Abelmoschus manihot* (L.) Medik. and *Taraxacum mongolicum* Hand.-Mazz. [2,3,4].

*Taraxacum mongolicum* Hand.-Mazz., known as the dandelion [5], is a plant that could produce a variety of bioactive metabolites, such as phenolic acids, flavonoids, polysaccharides, and terpenoids. These metabolites exhibit various benefits, including antibacterial, anti-inflammatory, hepatoprotective, choleretic, antitumor, gastrointestinal protective, diuretic, and skin-improving activities [6]. In herbal medicinal practice, it is regarded as “the natural antibiotics” and “herbal empresses” [7]. Although extensive research has been conducted on its growth characteristics, pharmacological activities, and chemical composition [8,9,10,11,12], limited studies have focused on the molecular mechanisms of salt tolerance in *Taraxacum officinale* under salt stress. Previous studies have demonstrated that integrative metabolome and transcriptome analysis can reveal changes in metabolite profiles and differences in gene expression patterns among plant genotypes with varying levels of salt tolerance within the same species under saline conditions [13].

Bioinformatics tools can be used to integrate information from different omics layers to systematically explore complex regulatory networks in biological processes, and identify new components within metabolic pathways by leveraging existing knowledge of the pathway relationships in multi-omics datasets. This strategy has been widely applied across many plant species [14]. For example, an integrative transcriptomic and metabolomic study conducted in *L. bicolor* under salt stress revealed higher accumulation of caffeoyl shikimic acid and coniferin in both low (S1) and high (S3, S4) salinity treatments. Flavonoid contents decreased under the S1 treatment but increased under the S3 treatment [15]. The corresponding expression trends for the key biosynthetic genes were found to be similar to the accumulation trends of their corresponding metabolites. Under high-salinity condition, osmotic and oxidative stresses induced increases in organic solutes and flavonoids as important stress responses. Another example included is an integrative transcriptomic and metabolomic study conducted on two salt-tolerant *C. rigescens* varieties, which revealed 114 metabolites with significantly altered abundances after five days of salt exposure. Many metabolites associated with phenylalanine and phenylpropanoid metabolism showed more increase than decrease abundance in both varieties, whereas specific genes (including HCT, β-glucosidases, F5H) and the metabolite 4-hydroxycinnamic acid may account for some part of the observed differences [16]. Another report described a comparison of leaves collected from mature blueberry trees grown under non-stress (control), low (S1), medium (S2), high (S3), and very high (S4) salinity treatments, and the resulting differentially abundant metabolomes [17]. Over half of them were flavonoids, with salt-induced upregulation of four major classes of flavonoid biosynthetic genes, including 4CL, F3′5′H, and LAR, causing greater accumulation of flavonols, glycosides, flavans, proanthocyanidins, and anthocyanins in the entire leaf tissue [18]. Similar results were observed in a study on the barren-tolerant wild soybean genotype GS2; cotyledon salt tolerance is associated with enhanced mobilization of reserves (lipids, sugars, proteins) and amino acid transport to the embryo axis/root, alongside improved antioxidant capacity involving ascorbic acid and naringin synthesis [19].

Therefore, we utilized *T. mongolicum* ‘BINPU2’ (variety A, salt tolerant mutant) and ‘TANGHAI’ (variety B, wild resource) to integrate transcriptomic and metabolomic datasets, which would help unravel the underlying mechanisms of salt response and provide valuable insights for breeding programs of this plant.

## 2. Materials and Methods

### 2.1. Plant Materials

Salt-tolerant mutant variety A (*Taraxacum mongolicum cv.* “BINPU2”, salt tolerance: 0.5% NaCl) and local wild resource variety B (*Taraxacum mongolicum *cv. “TANGHAI”, salt tolerance: 0.3% NaCl) were used in this study. BINPU2 was cultivated via the callus tissue exposed to salt stress [1]. Since April 2023, *Taraxacum mongolicum *cv. “BINPU2” has been grown experimentally in the Hebei Province Experimental Field of the Institute of Coastal Agriculture under natural environmental conditions with routine management until autumn; *T. mongolicum *cv. TANGHAI grows naturally in Hebei province. Seeds with five true leaves were potted in a flowerpot that is three times taller than its base diameter (32 cm × 28 cm) containing perlite, vermiculite, and peat at a ratio of 2:1:1, seven replicates each treatment being planted with one seedling per pot; seeds were sown in flowerpots filled with these premixed ones and watered every two days with 150 mL NaCl solution at 0.7% concentration or deionized water (CK) and allowed to develop for approximately 25 days till plants were shown developing 5 true leaves. After that, healthy growth was observed after checking visually from the initial stage of leaf expansion. Twenty-one days before the beginning of the regimen applied for the nutrition based on the solution containing 0.7% concentration of NaCl, and changing it every second day for six weeks using the salt tolerance response-related NaCl, followed by quick freezing in liquid nitrogen storage in the −80 °C refrigerator ready to be used in further multiomics downstream. While the plant growing period from June to July environmental temperature is around 25 °C–32 °C, with relative humidity oscillating around 65%.

### 2.2. Measurement of Physiological Indicators

Two dandelion lines, A and B, were cultivated under the salt stress condition of 0.7%. The content of soluble sugar and malondialdehyde (MDA), as well as the activity of antioxidant enzyme in plants, were detected quantitatively. Leaf extraction was performed according to the instruction manual for soluble sugar assays (Shanghai Enzyme-linked Biotechnology Co., Ltd., Shanghai, China). Superoxide dismutase (SOD) was assayed using a colorimetric method with guaiacol as the substrate. Plant peroxidase (POD) activity was determined by the nitrotetrazolium Blue chloride (NBT) photoreduction method; while MDA content was detected by colorimetric assay.

### 2.3. Primary Metabolome Profiling

Dandelion leaves were harvested, freeze-dried, and ground to powder. The further extraction processes followed the procedure in test kits. The resulted extracts were analyzed by the UPLC-ESI-MS/MS system (Waters, Milford, MA, USA) with the ExionL™ AD module (https://sciex.com.cn/, accessed on 27 April 2025) and Applied Biosystems 4500 Q TRAP (Applied Biosystems, Carlsbad, CA, USA, https://sciex.com.cn/, accessed on 27 April 2025) mass spectrometer. QC samples for the reproducibility check of the analytical results were also prepared by triplicate mixtures of all the samples. Three biological replications were performed in the first-level metabolomics experiment. The metabolic identification was conducted using our self-assembled database plus MWDB Metware database from Metware Biotechnology Co., Ltd. (Wuhan, China).

### 2.4. Transcriptome Profiling

We used electrophoresis on a 1% agarose gel to check the quality of RNA and any potential contamination. Purity was assessed using a NanoPhotometer^®^ spectrophotometer (IMPLEN, Westlake Village, CA, USA), and RNA concentration was determined with the Qubit^®^ RNA Assay Kit on a Qubit^®^ 2.0 Fluorometer (Life Technologies, Carlsbad, CA, USA), according to the manufacturer’s instructions. Sample integrity was evaluated using the Bioanalyzer 2100 Platform (Agilent Technologies, Santa Clara, CA, USA) with the included RNA Nano 6000 kit included (Agilent Technologies, CA, USA). Following sequencing, read data were aligned to the dandelion reference genome (GWHBCHG00000000.genome.fasta.gz; https://ngdc.cncb.ac.cn/gwh/Assembly/19733/show, accessed on 27 April 2025) using feature Counts v1.6.2/StringTie v1.3.4d. FPKM values obtained from this alignment were regarded as an indicator of gene expression levels. Then, between group differences in gene expression were analyzed using DESeq 2 v1.22.1 or edgeR v3.24.3 software. All these bioinformatic programs take multiple test correction into account with Benjamini–Hochberg method. Significant genes are chosen if |log_2_FoldChange| ≥ 1 and FDR < 0.05 [20]. Bioinformatic tools have been already described elsewhere [21]. Validation of transcript level by qRT-PCR has also been performed. Alternative splicing events, including five types (SE: skipped exon; RI: retained intron; MXE: mutually exclusive exons; A5SS: alternate 5′ splice site; A3SS: alternate 3′ splice site) [22] were examined by rMATS (v3.1.0).

### 2.5. Combined Profiling of Metabolome and Transcriptome Profiling

The same samples were used for both the metabolome and transcriptome analyses and the combined profile. Results of different types of metabolite analysis merged with those of different gene analyses were mapped onto KEGG pathway diagrams simultaneously in order to examine the relationships between genes and metabolites. Relations between gene products and metabolites based on orthologous genes downloaded from the KEGG GENES database were networked. The correlation value (Pearson) between each gene and metabolite was calculated using R’s correlation function, Subsequently, trend evaluation plots, correlation cluster maps, correlation network diagrams, and canonical correlation analysis were performed when correlations exceeded 0.80 and the *p*-value was below 0.05.

### 2.6. Data Analysis

Utilize Microsoft Excel 2019 (Excel) to draw Student’s *t* test results of significant differences between different samples (*p* < 0.05); use the relative software package of IGV (http://software.broadinstitute.org/software/igv/download, accessed on 27 April 2025) for drawing principal component analysis results, Venn diagrams, correlation heatmaps, clustering heatmaps, and volcano plots.

## 3. Results

### 3.1. Salt Stress Affected Physiological Index

#### 3.1.1. Changes in Soluble Sugar and MDA Content in Leaves of Varieties A and B Under Salt Stress

Soluble sugars are important osmotic regulation substances and have a good effect on reducing cell damage caused by environmental stress and increasing plant salt tolerance. MDA content reflects the degree of damage to the cellular membrane by oxidative stress. The effect of saline conditions on the two dandelion varieties (A and B) was studied through the determination of soluble sugar and MDA contents after they were treated with a saline environment. The results are shown in Figure 1.

As can be seen from Figure 1, during the whole process, the soluble sugar content of variety A was higher than that of variety B. The maximum value of the soluble sugar content of variety A was at 48 h after being treated with saline water, reaching 1.57 mg·g^−1^, and then it slowly decreased but remained at a high level. However, the soluble sugar content of variety B increased slowly for the first 36 h, and its maximum value reached 1.19 mg·g^−1^, which was significantly lower than that of variety A.

Notably, both varieties showed upward trends in MDA content, and in variety A it was substantially lower than in variety B. These results indicate that salt stress induced the accumulation of both soluble sugars and MDA. The superior salt resistance of variety A may be attributed to its ability to rapidly accumulate soluble sugars under stress conditions while more effectively inhibiting MDA production, thereby enhancing cellular adaptability to saline environments compared to variety B.

#### 3.1.2. Changes in Leaf Active Oxygen Enzyme Scavenging System of Varieties A and B Under Salt Stress

The effect of salt stress on SOD, POD, and CAT activities in the two dandelion varieties are shown in Figure 2. The results demonstrated that SOD, POD, and CAT had changed very irregularly throughout the whole experiment period; however, they were always much higher for variety A than those for B, and we could clearly see that their enzyme activity trends would raise firstly and then descend along the periods of time. Compared with B, variety A showed a very obvious increasing part of this reaction trend, which suggested its better resisting salinity. Specifically, at the 48 h salt treatment time point, the maximum enzyme activity levels in variety A were over 80% higher than those in variety B.

As part of the plant antioxidative defense mechanisms of plants—superoxide dismutase could capture the superoxide anion, peroxidase and catalase worked together to eliminate H_2_O_2_—the more protecting enzyme reactions would greatly inhibit the excessive reactive oxygen accumulation too much, with less oxidative damage, and maintain the steady state of cells—the former also reflects better salt-tolerating adaptation among different varieties.

### 3.2. Changes in Primary Metabolome in Variety A and B

Differences were analyzed between salt-treated samples (AS, BS) and their respective controls (ACK for Variety A and BCK for Variety B). A total of 1121 differentially accumulated metabolites were completely identified (Figure 3A, Appendix A). PCA and Pearson’s correlation analysis of the amounts of different metabolites emphasize significant metabolic variation across sample groups. ACK and AS had a better separation than BCK and BS; this suggests that variety A or variety B has varying degrees of salt tolerance or salt response mechanisms (Figure 3B). Heatmap analyses also showed a clear difference between the ACK vs. AS and the BCK vs. BS groups with a significant distinction between the two varieties (Appendix A).

Subsequently, the DAMs were screened out: 222 DAMs under salt stress conditions in variety A were identified (Figure 4(A-1), Appendix A). A KEGG analysis indicated that these DAMs are mainly involved in arginine biosynthesis and pyruvate metabolism. Classification according to DAM categories reveals three major categories: metabolism (76.2%), genetic information processing (4.76%), and environmental information processing (19.04%). Pathway category classification shows that they are involved in metabolic pathways (46, 73.02%), biosynthesis of secondary metabolites (32, 50.79%), biosynthesis of amino acids (10, 15.87%), and ABC transporters (10, 15.87%), as shown in Figure 4(B-2). For variety B, 147 metabolic contents were shown to be changed significantly under similar conditions (Figure 4(A-2), Appendix A).

A KEGG analysis demonstrated a prominent association between the DAMs and galactose metabolism and the pentose phosphate pathway (Figure 4(C-1)). Similar to variety A, these DAMs are classified into metabolism (70.59%), genetic information processing (3.92%), and environmental information processing (25.49%). These DAMs are distributed over pathways including metabolic pathways (43, 84.31%), biosynthesis of secondary metabolites (25, 49.02%), and ABC transporters (13, 25.49%), as shown in Figure 4(C-2). Therefore, we found that salt stress affects some crucial metabolic pathways essential for cellular growth and viability by altering the expression levels of ABC transporters in both varieties. Furthermore, the results obtained from the analysis of variety A seems more robust when compared to that of variety B. Together, these results highlight how significantly salt stress can affect the primary metabolome present in the leaves of the dandelions.

To summarize, under salt stress, the differential metabolites of the dandelion were mainly involved in three parts: metabolism, genetic information, and environmental information.

### 3.3. Salt Stress Triggers Transcriptional Reprogramming of Varieties A and B

To verify whether the changes to the metabolites in the dandelion leaves were related to the changed genes, we first compared the transcriptomes of ACK and AS leaves and BCK and BS leaves, respectively. The box plot FPKM (fragments per kilobase of transcript per million mapped fragments), FPKM density distribution, and Pearson correlation between biological replicates are shown in Appendix A, which verified the reliability of our data sets (Appendix A).

Based on the alignment outcome using Cufflinks software (Version 2.0) to reconstruct the transcripts, we annotated the genomic information of the new transcripts. According to the variable splicing results (Table 1), among five kinds of different splicing event model results, the SE (exon skipping) model accounted for the most differential alternative splicing events: it accounted for 51.5% (EventNum.JC) and 51.5% (EventNum.JCEC). However, A5SS (alternative 5′splice site) had the smallest ratio of differential alternative splicing, accounting for only 7.5% (EventNum.JC) and 7.6% (EventNum.JCEC).

From the heatmap and volcano plot in variety A, 59% of the 2468 DEGs were upregulated while the other 41% were downregulated (Appendix A). Among the top ten most significantly upregulated genes, two—GAA6 (TbA02G112480) and SNAK2 (novel.4196)—are gibberellin-regulatory proteins that participate in stress resistance, one—BGL44 (TbA05G102660)—plays an important role in glycosidic bond hydrolysis, PRF1 (TbA05G002320), encoding a 36.4 kDa proline-rich protein, has functions related to stress resistance; and SLAH3 (TbA05G105240) and CAHX (TbA06G066050) play significant roles in maintaining cell homeostasis in plants.

Among the top ten most significantly downregulated genes in variety A, PCO2 (TbA02G112810) activates hypoxia responses, SUS2 (TbA05G018990) can be commonly detected in plants and participates in sucrose breakdown and synthesis to produce UDP glucose and fructose, providing the UDP glucose precursors needed by the plant during the biosynthesis of cell wall polymers and starch, EGY3 (TbA02G006500), which encodes a zinc metalloprotease, may have a function involving hypocotyl elongation, and ERD7 (TbA06G093080) and NLTPT (bA08G032570) play crucial roles in plant defense responses under stressful conditions (Appendix A).

The volcano plot and the differential heatmap revealed that 57.7% of the total 3238 DEGs in variety B were upregulated, while 42.3% were downregulated (Appendix A). Among the top ten significantly upregulated genes, GASA6 (TbA02G112480) is an important gene for integrating adverse signals and increasing plant resistance, ASOL (TbA05G029800) is important for substance metabolism in plants, AB19A (TbA01G125120) and PTR1 (TbA08G041970) are related to plant growth and development, and PRF1 (TbA05G002320) and XTR6 (novel.9983) are involved in the metabolisms needed to maintain normal physiological functions.

Among the top ten significantly downregulated genes in variety B, GPX6 (TbA06G099400) encodes glutathione peroxidase 6 and participates in regulating cellular redox signal transduction and scavenging free radicals to improve stress tolerance, and LOX21 (TbA08G029470) and PSKR1 (TbA06G100780) participate in plant salt stress defense responses and growth and development regulation. Enzymes 3MAT (TbA05G034520) and 3MAT (TbA05G034550) participate in not only fatty acid synthesis and degradation but many other metabolic pathways needed to maintain normal physiological functions in living organisms. Y1765 (TbA07G014060), an LRR receptor-like serine/threonine protein kinase, regulates metabolism, cell activity, cell death, and normal immune system and nervous system activities as an energy mediator (Appendix A).

After sorting the differentially annotated results between varieties A and B’s transcriptomes from high to low by the DEGs, a functional enrich analysis was conducted on the first twenty significant DEGs. The results were closely associated with biotic/abiotic stresses. Among them, the maximum-fold-change ratio of GAA6 (encoding a gibberellin-related protein) between the two varieties reached 12.3 (*p* < 0.01). As a key node participating in the regulation network of plant defense responses and ROS homeostasis, the GAA6 gene could be chosen for subsequent verification and further study of its underlying molecular mechanism.

### 3.4. KEGG Pathways of the DEGs

The biological functions of the DEGs were analyzed further by KEGG pathway analysis and KOG annotation. Variety A under saline treatment (namely, ACK vs. AS) were assigned to 121 pathways in the KEGG database, and a KEGG analysis showed that the DEGs were mainly related to secondary metabolism biosynthesis (ko01100), phenylpropanoid biosynthesis (ko00940), and photosynthesis–antenna proteins (ko00500) (Figure 5A). The KOG results were involved in general function prediction only, post-translation modification, protein turnover, chaperones, and signal transduction mechanisms (Figure 5B). Variety B under saline treatment (namely, ACK vs. AS) were assigned to 124 pathways in the KEGG database, and a KEGG analysis showed that the DEGs were primarily enriched in plant pathogen interaction, phenylpropanoid biosynthesis, and photosynthesis–antenna protein (Figure 5C). The KOG results showed the relationship with general function prediction only, signal transduction mechanisms, and secondary metabolite biosynthesis, transport, and catabolism (Figure 5D).

The functional enrichment result described the DEGs between variety A and B significantly enriched in the phenylpropanoid biosynthesis pathway under saline stress at a false discovery rate less than 0.05, which was likely connected with the medicinal property as a traditional Chinese medicine; thus, salt stress may affect its characteristic pharmacological components’ biosynthesis by regulating the biochemical process.

A total of 181 TFs being uniquely expressed by dandelion varieties A and B represented 36 and 37 TF families. Among them, one special big group was the AP2/ERF-ERF176 family important to response to salt stress (Figure 6).

### 3.5. Verification of RNA-Seq Data

To verify the accuracy of the transcriptome results, 12 genes changed most significantly and, related to adversity and coercion, were selected for the qRT-PCR test from variety A and B. Upregulated genes GASA6 (TbA02G112480), BGL44 (TbA05G102660), and PRF1 (TbA05G002320), and downregulated genes PCO2 (TbA02G112810), SUS2 (TbA05G018990), and NLTPT (TbA08G032570) are from variety A; upregulated genes GASA6 (TbA02G112480), ASOL (TbA05G29800), and AB19A (TbA01G125120), and downregulated genes GPX6 (TbA06G099400), CAHX (TbA08G029470), and PSKR1 (TbA06G100780) are from variety B. The results indicated that the expressions of all 12 genes were in agreement with the results from the transcriptome (Figure 7).

### 3.6. Integrative Metabolome and Transcriptome Analysis Revealed Key Metabolic Pathways Affected by Saline Stress

Salt stress affected the metabolome and transcriptome of the dandelion dramatically (Figure 3 and Figure 4). In order to further study the relationship between the expression level of mRNA and the content of metabolites, the log2 FC values of varieties A and B in integrated omics data sets were compared, and the substances whose Pearson’s correlation coefficient was greater than 0.80 and p value was less than 0.05 were mapped into a nine quadrant map (Appendix A). The differential expression was shown by different quarters: quarters 2 and 8 show the significant increase or decrease of mRNA but not metabolites; quarters 4 and 6 represent only the change of metabolic levels but not the difference of mRNA; quarter 5 indicates no difference for both RNA and metabolites; quarters 3 and 7 show the opposite trend of mRNA and metabolite changes. The same comparison was performed with variety B (Figure 8A,B); therefore we obtained a lack of coherence among the directions of the two omic changes after comparing the two kinds of omic datasets. This difference may be due to the post-transcriptional regulation, such as translation, modification, degradation, etc., especially when we consider that the difference between the expression of mRNA and the expression of metabolites in AS is bigger than BS, which means AS responds differently to salt stress.

In summary, our integrative omic studies found the significantly changed pathways loaded with differentially expressed metabolites and genes in AS and ACK using threshold *p* value of <0.05, namely “Metabolic pathways”, “Biosynthesis of secondary metabolites”, and “Plant hormone signal transduction” (Figure 9A). Similarly, a KEGG pathway analysis of two omic results emphasizes the leading role of “Biosynthesis of secondary metabolites” (Appendix A). Enriched pathways are also found when comparing varieties A and B, namely “Metabolic pathways”, “Biosynthesis of secondary metabolites”, and “Plant hormone signal transduction” (Figure 9B). Similar KEGG pathway analysis results were observed when comparing varieties B and C (Appendix A).

With the multi-omics analysis between varieties A and B, we plotted how an individual’s main metabolic pathway (biosynthesis of secondary metabolites) responded to salt stress.

As shown in Figure 10A, the simplified network diagram indicated that there was a positive interaction between the gene product and metabolite involved in the transcriptome–metabolome relationship of variety A. Xyloglucan 1,4-β-glucosidase (TbA03G083900) was the member of the glycoside hydrolase family with a 3-N terminal; it acted mainly on starch and sucrose metabolism and secondary metabolic biosynthesis and other metabolic processes. Its mRNA expression level was upregulated. Meanwhile, beta glucosidase46 (TbA02G022670), which belonged to the glycosyl hydrolase family 1, was also highly expressed. The synergistic effect contributed to the plant responding to salt stress. Beta-fructofuranosidase (TbA02G130220) belonged to the class of the glycosyl hydrolases family 32 (soluble isoenzyme I); its mRNA expression level had been downregulated, and this may be mediated by the increase of core protein xyloglucan endotransglucosidase. Xyloglucan endotransglucosidase would affect the accumulation of beta-glucosidase46 but not beta fructofuranosidase.

Another central gene, cinnamyl CoA reductase1(A) (TbA01G018550) encoded a protein belonging to the NAD-dependent isomerase/dehydratase family, having an epigenetic isomerase domain. It played an important role in phenylpropanoid biosynthesis, metabolic pathways, and the formation of secondary metabolites, especially regulating the downregulation of CCR1 at the mRNA level. Under the regulation of the above central genes, hydroxycinnamoyl CoA shikimate—the substance engaged in phenylpropanoid and flavonoid biosynthesis—showed high mRNA expression levels due to the stimulation of salt stress. However, ferulic acid 5-hydroxylase 1 and coumarate CoA ligase 1 (TbA07G106840), a member of ubiquinone and other terpenoid quinone biosynthesis, were both downregulated due to the fact that, under the regulation of the central gene, the terpenoid metabolite accumulated less after being stimulated by salt stress.

As illustrated in Figure 10B, the simplified network diagram for variety B revealed that the core gene beta-glucosidase 46 (TbA02G022700), an isoform X2 of beta-glucosidase 18-like from the glycosyl hydrolase family 1, was upregulated, playing a main function in cyanoamino acid metabolism and starch and sucrose metabolism, whose mRNA expression level had been upregulated. Beta-glucosidase 17 (TbA03G023930), a member of the same glycosyl hydrolase family as the previous one, also significantly contributed to cyanoamino acid metabolism and starch and sucrose metabolism; meanwhile, beta-glucosidase 18 (TbA02G022670), a member of the glycosyl hydrolase family 1, was primarily concerned with these three metabolic processes. They both showed enhanced mRNA expression levels under the regulation of the core gene β-glucosidase 46. These changes in their mRNA expression levels may be regulated by increased activity of the glycosyl hydrolases participating in cyanoacetic acid metabolism under the stimulus of salt stress.

In conclusion, the integrated analysis of metabolomics and transcriptomics revealed that variety A and variety B had significantly changed in terpenoid metabolic pathway related metabolites (ubiquinone and other terpenoid quinones biosynthetic pathway), pyruvate metabolites with salt stress via mRNA, and/or metabolite level change induced by some important enzymes (glycoside hydrolase family, adenylate esterases family, and P450 cytochrome family).

## 4. Discussion

Soil salinization greatly limits crop production, and food security has become a serious problem in agricultural development worldwide [23]. The use of salt tolerant plant species would be a much more economical way to develop the utilization of saline–alkali lands [24]. The dandelion is one of the important medicinal plants which has been widely cultivated in China. It could grow well even if soil containing salt. In our study, we found different responses between two dandelion varieties affected by salt to elucidate the biosynthetic mechanisms how they grow normally under harsher conditions (saline) [25].

The plants’ ability of tuning their own metabolism helps them adapt to fluctuating environmental or abnormal metabolic states [26,27]. When interrupting one pathway, the whole metabolism network will be altered. Coordination of multi-gene and metabolite expression can provide a beneficial effect on fitness through disturbing the whole metabolic network after breaking the original balance [28,29]. Our data indicated that salt pressure not only disturbed a series of different kinds of metabolites but induced wide alternation of terpenoid biogenesis, especially ubiquinone and other terpenoid quinones, and the alterations were supported by gene expression and/or metabolite changes involved with special enzymes and TFs.

### 4.1. Glycoside Hydrolase Family

Sucrose and starch are major reserve carbohydrates in plants, and they have been well studied for energy storage and metabolism [30]. However, about 15% of angiosperms utilize fructans—a fructose polymer formed from sucrose by β(2-1) or β(2-6) bonds—as an alternative carbon store and stress adapter [31]. The enzymes involved in the degradation of fructan belong to the superfamily glycoside hydrolase (GH), which plays an important role in many biological processes, like cell wall remodeling, carbohydrate metabolism, defense response, and signaling [32]. GHs have already been classified into more than 100 families based on their structural relationship. Fructan-degrading enzymes include members of family BGALs that catalyze β-galactosyl residues at the non-reducing ends of carbohydrates, galactolipids, and glycoproteins. BGALs are distributed ubiquitously among different plant species, and participate in different important functions, like cell wall dynamics [33].

The results from the transcriptome and metabolome analysis revealed differential gene expression between the two varieties of dandelion under study. The transcriptomic results indicated changes in the abundance level corresponding to specific metabolic intermediates depending upon the treatment groups compared with each other; however, the metabolites were highly influenced by salt stress. The comparison of variety A and B transcriptomes gave us insight into how the stress resistance mechanism was influenced by salt stress in terms of change in expression levels of important enzyme-coding genes in the glycoside hydrolase family. Gene TbA03G083900 coding for xylan 1,4-β-glucosidase and gene TbA02G022670 coding for β-glucosidase 46 were expressed higher under salinity conditions, as reported previously. Similarly, we identified some new genes upregulated after salt-stress treatment, such as variety B TBGA0011190 encoding beta-fructofuranosidase and TBGA00124891 codes for xylan 1,4-beta-D-glucosidase, and both of them found significant increase in their transcript number during salt-stress condition. Such an analysis demonstrated the important role of glycoside hydrolases during plant response against abiotic stresses. Given the above discussion, glycoside hydrolase related genes can be identified as key areas for future research on salt tolerance in dandelions.

### 4.2. Adenylate Esterases Family

Adenylate is mainly AMP and has several functions. Adenosine monophosphate (AMP) plays a vital part in numerous processes regulated by adenylate cyclase, and participates in the promotion of plant growth and the development and resistance to stress [34]. Through reactions catalyzed by adenylate cyclase, adenosine monophosphate can be converted into various substances with biological activity, which regulate cell function or state.

In addition, there are also widespread relationships between AMP and different proteins. For instance, AMP combines with other proteins to form complexes, changing the protein’s functional activity, so as to play an essential role in all kinds of life [35]. It also connects with some other types of molecules, such as sugar and fat, to form a complicated metabolism to maintain life in organisms and provide normal physiological conditions [36]. There were changes concerning the transcription expression of the adenylate esterase family under salt pressure by transcriptome analysis (Figure 4(B-2,C-2)). The results showed convincing proof that this specific dandelion was capable of enduring stressing environments. Metabolism changes reflected modification in carbohydrate content caused directly by plants reacting against stressful situations when compared with the CK group. The interaction change factors related to AMP and the relative molecular substances resulted from comparisons with control, and indicated its special function and meaning in responding to our environment differently. However, the specific functions of the adenosine monophosphate DEGs that are related to salt tolerance still need to be verified.

### 4.3. P450 Cytochrome Family

As an enzyme superfamily encoded by a set of genes, CYP450s is the first to be found. CYP450s widely exists in many kinds of plant tissues and organs with various catalytic functions, and is essential to some basic or non-basic metabolism reactions which are very important for plants’ lives [37]. The 11 family clusters, including 63 gene families, have been classified in the plants.

It has been proven that the regulation effect on the biosynthesis of terpenoids: TaCYP71C1 reduced under salt stress by decreasing expression, which lead to poor salt tolerance; overexpression of TaCYP71C1 made wheat more sensitive to salt, confirming its negative regulatory function under saline conditions [38]. We concluded the significant role of TaCYPs genes in resisting environmental stress. So we provide a good basis for our further studies on whether TaCYPs would play such regulatory effects in other abiotic stresses, and whether it will be beneficial for crops in breeding. Meanwhile, cytochrome P450s regulates the defense mechanism against the biotic and abiotic stresses of plants by affecting phytohormones and secondary metabolites, enhancing ROS scavenging ability, and improving biochemical and molecular defenses. Cytochrome P450s have always been the first choice to modify because they can improve stress resistance [39]. In this paper, ferulic acid 5-hydroxylase 1(A) (TbA08G044760) belonging to the CYP450 gene family was downregulated under salt pressure, which has been supported by the literature.

StCYP230, StCYP94D2, and StCYP86B2 showed similar functions according to the gene phylogenetic tree. The decreased degradation of chlorophylls, increased proline level, and oxidative enzymes (peroxidase; POD), and they were reflected as an enhanced abiotic stress tolerance in a particular genotype [40]. After genome-wide identification in potato genomes by homology modeling, 253 StCYP family genes were observed and one member of them was inferred to be involved in response towards water deficiency stress through ABA-dependent pathway using their sequence information. Further RNA-seq data analysis of pigmented tuber flesh of three potato clones at different time points identified multiple members from the *S. tuberosum* CYP86 genes possibly functioning in tuber flavonoid biosynthesis process. Using GFAnno software (Version 1.4) with the help of RNA-seq data, it was predicted that seven of the StCYP genes, like two StC4H, three StF3′H, one StFNS II, and one StF3′5′H, are the major players responsible for functional flavonoid biosynthetic pathways, and five of the CYP 71 clan correlated well with the flavonoid metabolism [41]. Our observation indicated increased flavonoid biosynthesis under salt stress in the dandelion consistent with our previous study, and provided more evidence which can sustain plant adaptation in saline environments. It is obvious that biological processes are multifaceted, so we need an integrated view, not isolated perspectives, of such multilevel omics results to understand macroscopic developmental phenomenon. Such interlevel integrations allow us to cross validate these different kinds of data thus enabling a panoramic perspective about the biology of interest. Cross-talk establishment over various levels along with its annotations facilitate the comprehensive assessment of biomolecule function and regulatory network across levels by applying functional annotation methods, metabolic pathway enrichment studies, and others.

All these approaches aid our ability to gain insight into the broader patterns governing the biochemical transformations. Thus multi-omics analysis concluded that there occurred modification in the expression of genes with great significance, while induction in huge amounts with reference to metabolic shift towards starch, soluble sugar, amino acid, and lipids under the effect of salt stress condition.

## 5. Conclusions

According to the results of metabolomics and transcriptomics analysis, the potential salt-response mechanisms in the dandelion under saline stress were explored. The results showed that there are significant differences in metabolite profiles and gene expression patterns between dandelion genotypes with different levels of salt tolerance during salt stress. While variation was detected across all sample groups, the separation between ACK and AS was more pronounced than that between BCK and BS. Notably, the number of DEGs between genotype A and B were opposite for the two samples. Some salt-stress pathways, including secondary metabolism biosynthesis and phenylpropanoids biosynthesis, were enriched. All the above suggested that the dandelion had a different responding mechanism to salt stimulus between these two types, and dandelion A (BINPU2) would have more tolerance to salt stress than dandelion B (TANGHAI). Both dandelions will experience a dramatic change of terpenoids-, ubiquinone-, and pyruvate-related metabolic activities because of various enzyme action from glycoside hydrolases, adenylate esterases, and P450 cytochromes in response to salt conditions at transcriptional and/or metabolic levels.

## Figures and Tables

**Figure 1 cimb-47-00449-f001:**
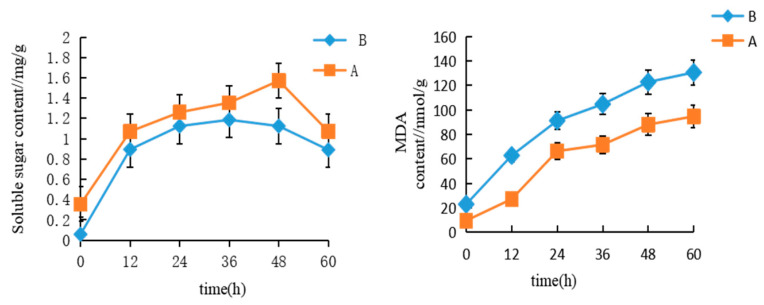
Performance of two dandelions under different NaCl treatments. Orange represents dandelion A (BINPU2), blue represents dandelion B (TANGHAI).

**Figure 2 cimb-47-00449-f002:**
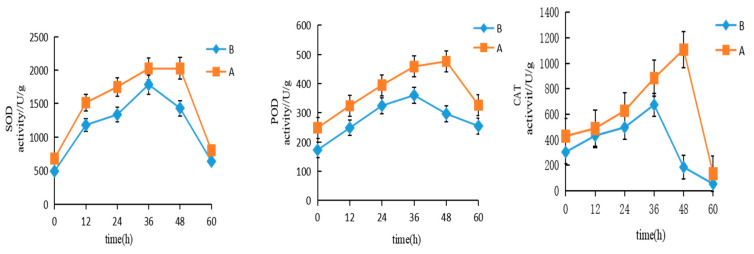
The antioxidant enzyme activity of two dandelions under different NaCl treatments. Orange represents dandelion A (BINPU2), blue represents dandelion B (TANGHAI).

**Figure 3 cimb-47-00449-f003:**
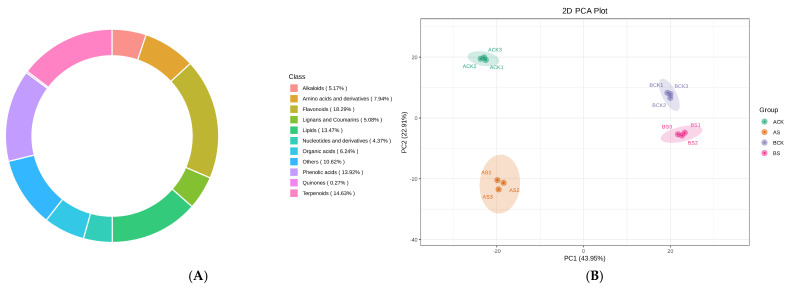
(**A**) Component analysis of identified metabolites in leaves of varieties A and B. (**B**) Principal component analysis of identified metabolites in leaves of varieties A and B.

**Figure 4 cimb-47-00449-f004:**
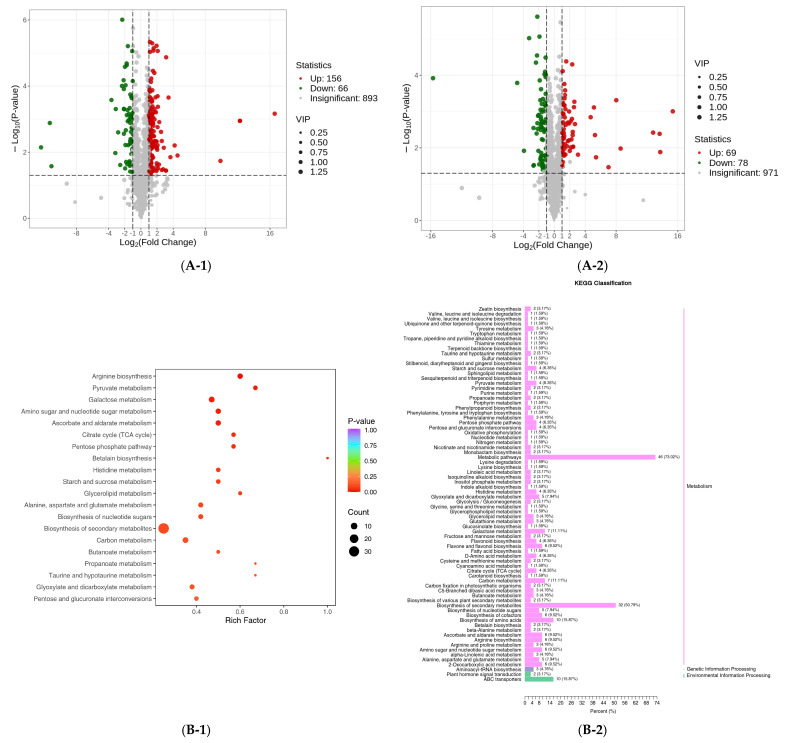
Summary of the main metabolome and DAMs of varieties A and B. (**A-1**) The volcano map of the different metabolites of variety A. (**A-2**) The volcano map of the different metabolites of variety B. (**B-1**) KEGG pathway finding between ACK and AS groups. (**B-2**) Composition and proportion of the DAMs of ACK and AS groups. (**C-1**) KEGG pathway finding between BCK and BS group. (**C-2**) Composition and proportion of the DAMs of BCK and BS groups.

**Figure 5 cimb-47-00449-f005:**
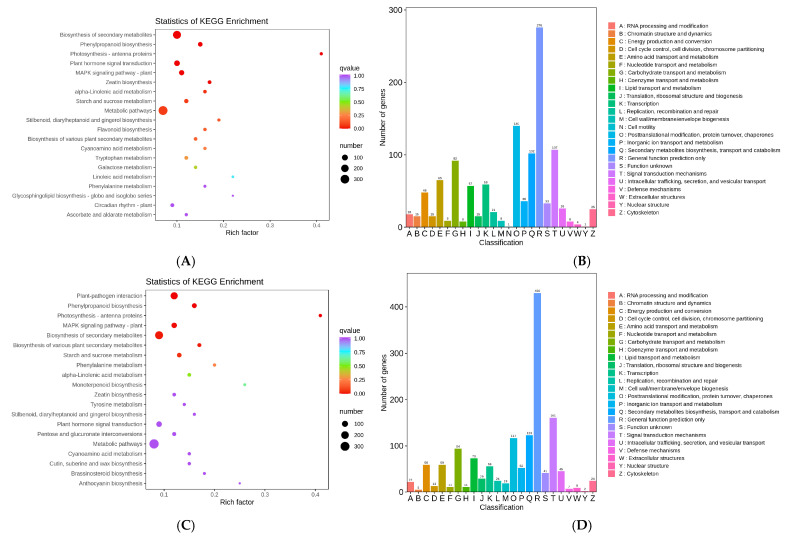
Summary of primary transcription in variety A and B. (**A**) KEGG pathway enrichment between ACK and AS. (**B**) KOG annotation between ACK and AS. (**C**) KEGG pathway enrichment between BCK and BS. (**D**) KOG annotation between BCK and BS.

**Figure 6 cimb-47-00449-f006:**
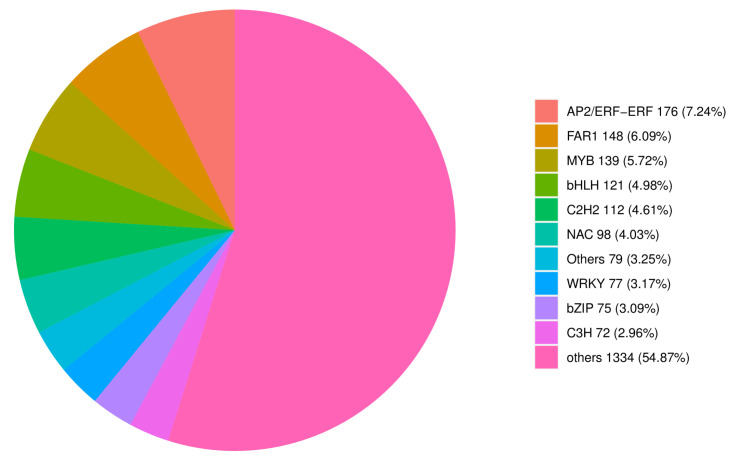
Family classification of transcription factors for new transcript information.

**Figure 7 cimb-47-00449-f007:**
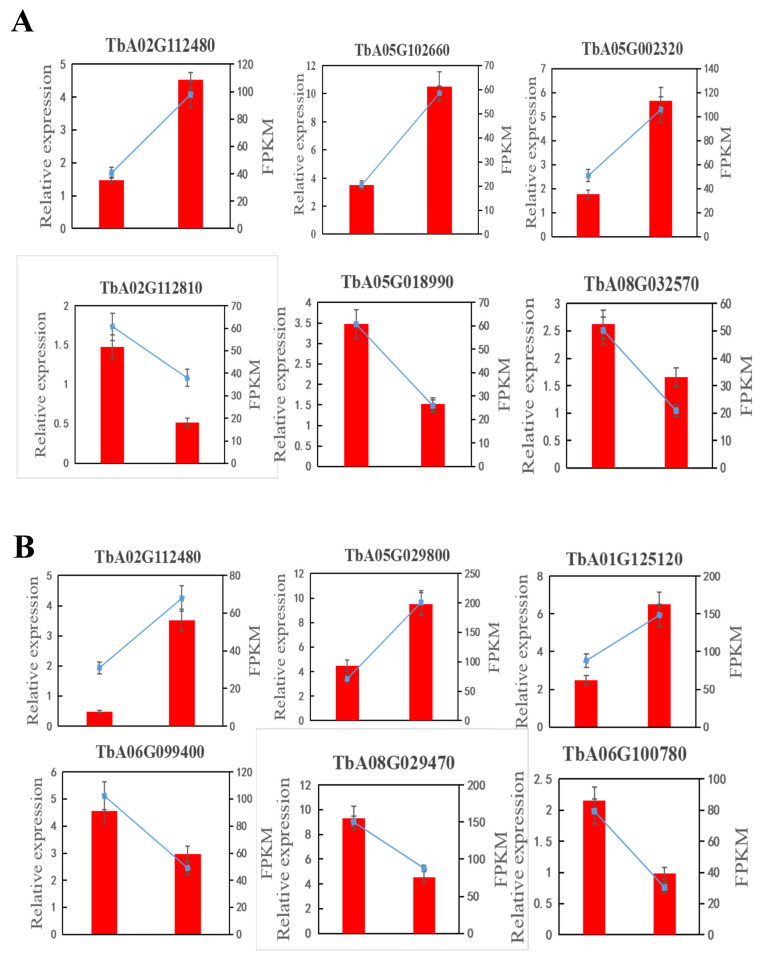
Quantitative real-time PCR (qRT-PCR) validation of the RNA-Seq results: (**A**) dandelion A (BINPU2), (**B**) dandelion B (TANGHAI). The qRT-PCR results are shown in bar chart. Each bar represents the average of three biological replicates (mean ± SD). The value obtained from WT was set at 1. Fragments per kilobase of exon per million fragments of mapped reads (FPKM) are shown in the line chart.

**Figure 8 cimb-47-00449-f008:**
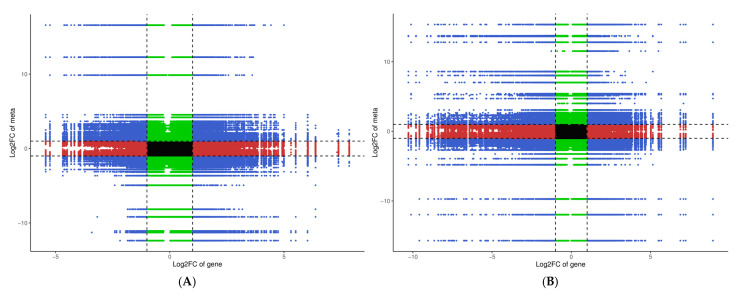
Correlation analysis of nine quadrant diagram of varieties A and B: (**A**) dandelion A (BINPU2), (**B**) dandelion B (TANGHAI). Note: Use black dashed lines to divide from left to right and from top to bottom into 1–9 quadrants, The horizontal axis represents the log2FC of genes, and the vertical axis represents the log2FC of metabolites.

**Figure 9 cimb-47-00449-f009:**
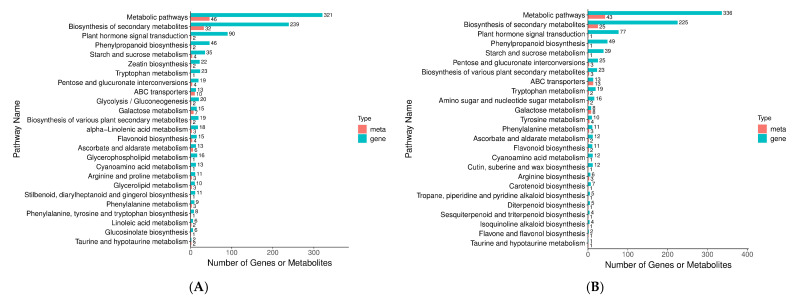
Summary of primary transcription and metabolome in varieties A and B: (**A**) KEGG pathway enrichment between ACK and AS, (**B**) KEGG pathway enrichment between BCK and BS. Note: Red indicates upregulation of genes or gene products or metabolites, while green indicates downregulation of genes or gene products or metabolites.

**Figure 10 cimb-47-00449-f010:**
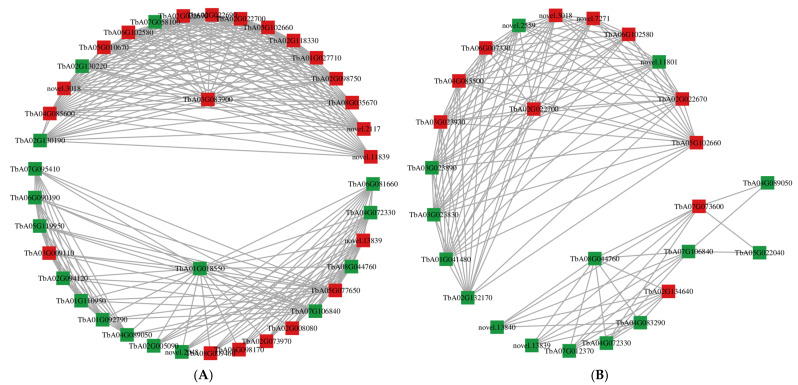
Network relationship between gene products and metabolites in variety A and B. (**A**) Interrelationship between transcriptome and metabolome in variety A. (**B**) Interrelationship between transcriptome and metabolome in variety B. Red represents upregulated genes, green represents downregulated genes.

**Table 1 cimb-47-00449-t001:** Alternative splicing.

AS_Type	EventNum.JC	SigEventNum.JC	EventNum.JCEC	SigEventNum.JCEC
A3SS	950	96 (62;34)	950	101 (65;36)
A5SS	426	29 (17;12)	429	32 (20;12)
MXE	199	26 (14;12)	201	32 (17;15)
RI	1154	83 (59;24)	1155	83 (60;23)
SE	2900	131 (32;99)	2924	141 (39;102)

## Data Availability

Data are contained within the article and the Appendix A.

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
