# Peer review of "Preliminary Analysis of the Salt-Tolerance Mechanisms of Different Varieties of Dandelion (Taraxacum mongolicum Hand.-Mazz.) Under Salt Stress"

_cimb, 2025, doi:10.3390/cimb47060449_

Round 1

Reviewer 1 Report

Comments and Suggestions for Authors

Exploring the metabolites and associated pathways is improtant for improving the salt tolerance of dandelion.  Although some results were found in this paper, it still needs to be improved. 

  1. Samples are not clearly introduced, especially wild resource B, salt tolerance or not.
  2. In experiment design, there is no control. Most importantly, there is only the leaves analyzed, why the root is not anlyzed. Generally, root is important for salt-tolerance.
  3. Figures and Tables are not good, should be improved. 
  4. The format of this paper should further improved.

Author Response

Comments 1:Samples are not clearly introduced, especially wild resource B, salt tolerance or not.

Response 1:Thank you for pointing this out. I/We agree with this comment. Therefore, I/we haveWe have clearly introduced the salt tolerance of wild resource B on the third line of page 4

Comments 2: In experiment design, there is no control. Most importantly, there is only the leaves analyzed, why the root is not anlyzed. Generally, root is important for salt-tolerance.

Response 2: This suggestion is highly targeted. Nevertheless, given that Taraxacum officinale(dandelion) serves as a significant medicinal and edible plant, with its leaves playing a pivotal role in both production and application, this article intends to conduct a focused exploration into the characteristics and mechanisms underlying its leaf salt tolerance.

Comments 3: Figures and Tables are not good, should be improved.

Response 3:Thank you for pointing this out.We have made improvements to the figures and tables.

Comments 4: The format of this paper should further improved

Response 4:Thank you for pointing this out.We have made improvements to The format of this paper.

Comments 5 Response to Comments on the Quality of English Language

Response 5:We have improved the Quality of English Language

Reviewer 2 Report

Comments and Suggestions for Authors

Review of “Comprehensive analysis of transcriptomics and metabolomics elucidates the mechanisms underlying salt-tolerance in dandelion (Taraxacum mongolicum Hand.-Mazz.)”

The study is relevant as it uses transcriptomics and metabolic analyses to address the slat-tolerant mechanism of a non-model medicinal plant. However, several issues must be addressed before being considered for publication.

Issues:

  1. All the scientific names should be italicized.
  2. The title should not include transcriptome and metabolomics, as this is common these days and does not add any value to the manuscript. The authors themselves did the transcriptome analysis of the same plant before. Please rephrase it.
  3. The introduction lacks citations in several instances; make sure to add references wherever applicable.
  4. I do not buy the author’s argument of making this hypothesis that they did this study as no work focused on the salt stress mechanism based on combined metabolomics and transcriptomics methods. Provide a proper hypothesis and list knowledge gaps.
  5. Provide the full names for MDA, ACK, BCKs, and all abbreviations in general.
  6. Why do the legends have Chinese legends in the axes of the figures?
  7. In qPCR experiments, the genes selected—how the expression for qPCR correlated with transcriptome data. Please introduce that.
  8. Again, there are not enough citations in the discussion.
  9. There are grammatical errors throughout the manuscript; please fix those.
  10. Figure outputs from transcriptome analyses are not adequately explained in the text.
  11. The manuscript is data-heavy without a detailed interpretation of the data.
  12. The conclusion fails to highlight the study's novel contributions or suggest future research directions.
Comments on the Quality of English Language

It must be improved.

Author Response

Comments 1: All the scientific names should be italicized.

Response 1: [Type your response here and mark your revisions in red] Thank you for pointing this out.We have made modifications

Comments 2:The title should not include transcriptome and metabolomics, as this is common these days and does not add any value to the manuscript. The authors themselves did the transcriptome analysis of the same plant before. Please rephrase it.

Response 2: Agree. We have changed The title to Preliminary analiysis of salt-tolerance Mechanisma of Different Varieties of Dandelion(Taraxacum mongolicum Hand.-Mazz.) under salt stress.

Comments 3: The introduction lacks citations in several instances; make sure to add references wherever applicable,Again, there are not enough citations in the discussion

Response 3:We have added references in the Introduction and  discussion

Comments 4: I do not buy the author’s argument of making this hypothesis that they did this study as no work focused on the salt stress mechanism based on combined metabolomics and transcriptomics methods. Provide a proper hypothesis and list knowledge gaps.

Response 4: However, there was no work focused on the salt stress mechanism based on combined metabolomics and transcriptomics methods although lots of researches had been done about its growth characteristic, pharmacological activity, and chemical composition[8-12].Change to Although extensive research has been conducted on its growth characteristics, pharmacological activities, and chemical composition [8-12], limited studies have focused on the molecular mechanisms of salt tolerance in Taraxacum officinale under salt stress

Comments 5: Provide the full names for MDA, ACK, BCKs, and all abbreviations in general.

Response 5: MDA: Malondialdehyde; POD: Peroxidase;SOD:Superoxide dismutase; ACK: Variety A control; BCK: Variety B control ;NBT: Nitrotetrazolium Blue chloride. The full names of other abbreviations have also been provided

Comments 6: Why do the legends have Chinese legends in the axes of the figures?

Response 6: The figure on page seven has been modified to an English legend

Comments 7:In qPCR experiments, the genes selected—how. Please introduce that.在qPCR

Response7: The relationship between qPCR expression and transcriptome data of selected genes is introduced on page 13

Comments 8:Figure outputs from transcriptome analyses are not adequately explained in the text.

Response8:Agree.We have provided a detailed introduction to the output of transcriptome analysis graphs

Comments 9:The manuscript is data-heavy without a detailed interpretation of the data.

Response9:Thank you very much for the reviewer's comments. Although the manuscript has a large amount of data, we focused on explaining the parts related to salt stress and did not provide detailed explanations for the remaining parts.

Comments 10:The conclusion fails to highlight the study's novel contributions or suggest future research directions.

Response10:Agree.We have added future research directions to the conclusion

Response11:There are grammatical errors throughout the manuscript; please fix those.

Response 11:We have fixed the grammar errors in the manuscript and highlighted them in red

Round 2

Reviewer 2 Report

Comments and Suggestions for Authors

The author's revisions are satisfactory.